# Spatial variation in housing construction material in low- and middle-income countries: A Bayesian spatial prediction model of a key infectious diseases risk factor and social determinant of health

Josh M. Colston[1,2]*, Bin Fang[3], Malena K. Nong[4], Pavel Chernyavskiy[2], Navya Annapareddy[5], Venkataraman Lakshmi[2], Margaret N. Kosek[1,2]

1 Department of Medicine, Division of Infectious Disease and International Health, School of Medicine, University of Virginia, Charlottesville, Virginia, United States of America, 2 Department of Public Health Sciences, University of Virginia School of Medicine, Charlottesville, Virginia, United States of America, 3 Department of Civil and Environmental Engineering, University of Virginia, Charlottesville, Virginia, United States of America, 4 College of Arts and Sciences, University of Virginia, Charlottesville, Virginia, United States of America, 5 School of Data Science, University of Virginia, Charlottesville, Virginia, United States of America

* josh.colston@virginia.edu

## Abstract

Housing infrastructure and quality is a major determinant of infectious disease risk and other health outcomes in regions where vector borne, waterborne and neglected tropical diseases are endemic. It is important to quantify the geographical distribution of improvements to dwelling components to identify and target resources towards populations at risk. This study aimed to model the sub-national spatial variation in housing materials using covariates with quasi-global coverage and use the resulting estimates to map predicted coverage across the world's low- and middle-income countries. Data on materials used in dwelling construction were sourced from nationally representative household surveys conducted since 2005. Materials used for construction of flooring, walls, and roofs were reclassified as improved or unimproved. Households lacking location information were georeferenced using a novel methodology. Environmental and demographic spatial covariates were extracted at those locations for use as model predictors. Integrated nested Laplace approximation models were fitted to obtain, and map predicted probabilities for each dwelling component. The dataset compiled included information from households in 283,000 clusters from 350 surveys. Low coverage of improved housing was predicted across the Sahel and southern Sahara regions of Africa, much of inland Amazonia, and areas of the Tibetan plateau. Coverage of improved roofs and walls was high in the Central Asia, East Asia and Pacific and Latin America and the Caribbean regions. Improvements in all three components, but most notably floors, was low in Sub-Saharan Africa. The strongest determinants of dwelling component quality related to urbanization and economic development, suggesting that programs should focus on supply-side interventions, providing resources for housing improvements directly to the populations that need them. These findings are made available to

**Data Availability Statement:** The human subject data used in this analysis were collected and owned by third parties, not by the authors. They are all publicly available and can be accessed from the following sources in the same manner that the authors accessed them: - Demographic and Health Surveys, Malaria Indicator Surveys, and AIDS Indicator Surveys - https://dhsprogram.com/ - Multiple Indicator Cluster Surveys - https://mics.unicef.org/surveys - Botswana Family Health Survey - https://microdata.statsbots.org.bw/index.php/catalog/9 - Encuesta Nicaragüense de Demografía y Salud - https://www.inide.gob.ni/Home/endesa - Encuesta Nacional de Salud y Nutrición - https://www.salud.gob.ec/encuesta-nacional-de-salud-y-nutricion-ensanut/ - Pesquisa Nacional de Demografia e Saúde da Criança e da Mulher - https://bvsms.saude.gov.br/bvs/pnds/ The data on the covariate predictors are available from the sources cited in Table 2. The authors do not have the right to share these datasets, but did not have any special access privileges that others would not have.

**Funding:** This research was supported financially by the National Institutes of Health's National Institute of Allergy and Infectious Diseases (grants 1K01AI168493-01A1 to JMC and 1R03AI151564-01 to MNK); the European Union under the HORIZON EUROPE Programme (Grant Agreement Number: 101137255 with sub-award to JMC); the Engineering in Medicine (EIM) funding program (to VL), the Department of Internal Medicine and the Division of Infectious Diseases and International Health at the University of Virginia. The funders played no role in the design and implementation of the study or the analysis and interpretation of the results.

**Competing interests:** The authors declare that no competing interests exist.

researchers as files that can be imported into a GIS for integration into relevant analyses to derive improved estimates of preventable health burdens attributed to housing.

## Introduction

The United Nations' Sustainable Development Goals (SDGs) include ambitious commitments to fight communicable diseases (target 3.3) and provide adequate, safe and affordable housing (target 11.1) throughout its member states [1]. Although they fall under separate goals, housing quality has long been recognized as a social determinant of health and epidemiological evidence is now elucidating the mechanisms by which this relationship operates [2]. Many endemic infectious diseases of global public health concern, including several named in SDG3, are transmitted within and between households with the majority of infections occurring while the susceptible individual is at home [3], and consequently features of the built peridomestic environment and infrastructure play a role in promoting or impeding the spread of pathogens and their insect vectors [4]. This is particularly true in tropical and rural regions of Africa, Asia and Latin America where numerous vector borne and neglected tropical diseases circulate and where dwellings are often constructed using locally available, naturally occurring materials and traditional techniques such as wattle and daub, dried or burnt bricks, adobe, woven reed or bamboo and thatch [4]. These construction methods often require great skill and community mobilization to implement and are adapted over generations to suit local climate, ecology and topography, however numerous disease-causing insects and microbes are also well adapted to take advantage of the ecological niches that such buildings provide [5,6].

Infants and young children are particularly vulnerable to the health effects of housing construction material due to the high proportion of time spent in the family dwelling and behaviors common to early life such as crawling or playing on the floor [7–9]. Floors that are finished with wood, tiles or cement may protect against transmission of some diarrhea-causing enteric pathogens compared to those made of packed earth or sand either because they are easier to clean, or because they are less hospitable to pathogen survival outside the host [9]. Finished floors have been associated with decreases of 0.89 in $Log_{10}$ *E. coli* contamination in Peru [10], 78%, 43% and 27% in soil-transmitted helminthiasis prevalence respectively in Mexican, Bangladeshi, and Kenyan children [11,12], and 9% for diarrheal disease risk, 11% for both enteric bacteria and enteric protozoa risk [8], and 17% for *Shigella* spp. infection probability in meta-analyses of children under 5 years across multiple LMIC surveillance sites [13]. Traditional roof material has also been shown to be associated with childhood diarrhea [14], even after adjusting for floor material [15]. Pooled analyses of household survey data from multiple countries have found associations of living in improved housing on numerous child health outcomes, including cognitive and social-emotional development [7], and nutritional status [16], in addition to malaria infection [17,18]. Additionally, there is evidence of increased acute respiratory illness (ARI) in children in Pakistan, with unimproved flooring increasing ARI risk by 18%, and unimproved walling materials also increasing the risk of ARI in children under the age of five [19]. These findings are supported by similar findings with different studies in India, Nigeria, and Lao PDR [20–22].

As childhood mortality continues to decline globally, becoming concentrated in subnational hotspots it will be increasingly necessary to target interventions ever more specifically both geographically and to particular causes [23]. Several household-level determinants of health have been mapped at continental or global scale using survey data and spatial

interpolation methods including water source and sanitation facility type [24], crowded living space [25], educational attainment [26], and relative wealth [27]. Tusting and colleagues have applied a similar approach to mapping houses built with finished materials across Sub-Saharan Africa for the years 2000 and 2015, defining such households as those having at least two out of three of the materials for the walls, roof and floor being finished, though they did not separate out these three components in their main analysis [28]. Building on these efforts, the aim of this study, a project of the Planetary Child Health & Enterics Observatory (Plan-EO, www.planeo.earth) [29], is to provide estimates of the coverage of improvements in each of these three dwelling components that can be used to more effectively target infectious disease control measures and other health interventions. To achieve this, we model the sub-national spatial variation in housing materials using covariates with quasi-global coverage and use the resulting estimates to map the predicted coverage across low- and middle-income countries (LMICs). The guiding hypothesis was that coverage of improved housing materials varies spatially as a function of environmental, and socio-demographic factors in a way that can be modelled using publicly available global datasets and state-of-the-art geostatistical methods.

## Materials and methods

### Objective and scope

The objective of this analysis was to estimate the percent coverage of each category of materials used in dwelling component construction at all locations throughout the world's LMICs (as defined by the Organisation for Economic Co-operation and Development [30], excluding those in Europe).

### Outcome variables

The categories of housing materials used in this analysis were those proposed by Florey and Taylor [18], who classify materials used for construction of flooring, walls, and roofs into natural, rudimentary, and finished types, and then further into improved and unimproved. Data relating to these variables were compiled from nationally representative, population-based household surveys with two-stage cluster-randomized sample designs such as the Demographic and Health Surveys (DHS) [31], the Multiple Indicator Cluster Surveys (MICS) [32] and others from sources provided in S1 Data. These programs collect information on coverage of health and development indicators and make the resulting microdata publicly available through their websites. All Standard DHSs, Malaria and AIDS Indicator Surveys (MIS and AIS) and MICS dating back to 2005 that collected information on housing material from any LMICs were included. For countries where no such surveys were available, either similar surveys from the 2000–2004 period or country-specific surveys were sourced where available. The unit of analysis was the household, and these were classified into three, mutually exclusive categories (natural, rudimentary, and finished) based on the housing material recorded by the survey interviewer for each of the three dwelling components (floors, walls, and roof) as shown in Table 1.

   **Georeferencing households.**   For this spatial analysis it was necessary to assign coordinates to each household representing its approximate location. Cluster-randomized surveys have a hierarchical design such that households are nested within clusters, the census enumeration areas that serve as the primary sampling unit, which are in turn nested within survey strata (sub-national region and urban/rural status). Typically, 25–30 households are sampled per cluster [33]. The DHS Program provides coordinates of the cluster centroids for most of the surveys they carry out [34] (though these are randomly "displaced"–systematically shifted up to a certain distance to preserve confidentiality [35]). However, these are not available for

**Table 1. Classification of construction materials for the three components of the dwelling used as three-category outcome variables (adapted from Florey and Taylor 2016 [18]).**

| Category | | Flooring | Walls | Roof |
|---|---|---|---|---|
| **Unimproved** | Natural | Earth, sand, dung etc. | Mud, sticks, cane, palm, tin, cardboard, paper, thatch, straw etc.<br>No walls | Grass, thatch, palm leaves, sod, straw etc.<br>No roof |
| | Rudimentary | Wood planks, palm, bamboo etc. | Bamboo, stone, or trunks with mud, uncovered adobe, plywood, cardboard, reused wood, unburnt bricks etc. | Palm, bamboo, wood planks, cardboard, tarpaulin, plastic etc. |
| **Improved** | Finished | Parquet or polished wood, vinyl or asphalt strips, cement, carpet etc. | Cement or cement blocks, stone with lime or cement, bricks, covered adobe, wood planks/shingles, burnt bricks etc. | Metal, wood, ceramic tiles, cement, shingles, slate etc. |

all clusters and surveys and equivalent coordinates have been made available only for a handful of MICS and no country-specific surveys. For this analysis, households were georeferenced to their displaced cluster centroid coordinates where available, otherwise their clusters were randomly assigned to populated settlement locations taken from the Humanitarian OpenStreetMap database [36] that fell within the same survey stratum (sub-national region and urban/rural status) with probability proportional to the population density of the settlement (extracted from the WorldPop [37] database at settlement coordinates). OpenStreetMap settlements were reclassified such that cities and towns were categorized as urban, and villages, hamlets, and isolated dwellings as rural. This novel cluster location assignment process was automated in ArcGIS Pro ModelBuilder [38] and Stata 18 [39].

## Covariates

A suite of time-static environmental and demographic spatial covariates available in raster format were compiled based on their hypothesized associations with the outcome variables. Definitions and sources of each covariate are shown in Table 2. Variable values were extracted at the georeferenced cluster locations in Python. In addition, time was calculated in continuous months since January 1st, 2005, based on the date of survey interview and log transformed. This choice was informed by the assumption that changes in material used for housing tend to be infrequent and unidirectional–i.e. households seldom change the material used in their construction and when they do, it is generally from unfinished to finished and not the other way. Countries were grouped into the six regions used for administrative purposes by the World Bank [40], and this categorical variable was also treated as a covariate so that, for countries with no available survey data, estimates would be based partly on regional averages.

## Analysis

To reduce the database size and computational demands, and to neutralize the issue of within-cluster correlation, one household with non-missing outcome value was randomly sampled per cluster and retained for analysis (this selection was done separately for each of the three outcomes). This also ensured that each sampled location had just one value for the outcome variable. Due to the computational demands of performing geospatial analysis at the global scale, we recoded all outcomes to be binary, by collapsing two of the response categories together ("rudimentary" was grouped with "natural") to give "improved" / "unimproved" response categories as shown in Table 1, and in a modification of the schema proposed by Florey and Taylor (those authors grouped rudimentary and finished walls and roofs into the improved category, but not floors, however we opted for a consistent categorization across components to facilitate comparison between outcome variables [18]).

**Table 2. Definitions and sources of variables included as covariate predictors in the model.**

| Variable | Definition | Units/ Categories[1] | Source |
|---|---|---|---|
| **Accessibility to cities** | Travel time to nearest settlement of >50,000 inhabitants. | Minutes | MAP [46] |
| **Aridity index** | Mean annual precipitation / Mean annual reference evapotranspiration, 1970–2000. | Ratio | CGIAR-CSI [47] |
| **Climate zone** | First level Köppen-Geiger climate classification. | Tropical; arid; temperate; cold; polar | Beck et al. 2018 [48] |
| **Cropland areas** | Proportion of land given over to cropland, 2000. | Proportion | CIESIN [49] |
| **Distance to major river** | Distance to major perennial river (derived from rivers and lakes centerlines database). | Decimal degrees | Natural Earth [50] |
| **Elevation** | Elevation above sea level. | Meters | NOAA [51] |
| **Economic development** | Sub-national unit-level Gross Domestic Production (GDP) per capita, 2015 | Constant 2011 int. USD | Kummu et al. 2018 [52] |
| **Enhanced Vegetation Index** | Vegetation greenness corrected for atmospheric conditions and canopy background noise. | Ratio | USGS [53] |
| **Growing season length** | Reference length of annual agricultural growing period (baseline period 1961–1990). | Days | FAO, IIASA [54] |
| **Human development** | Sub-national unit-level Human Development Index (HDI), 2015 | Scale from 0 to 1 | Kummu et al. 2018 [52] |
| **Human Footprint Index** | Human Influence Index (HII) normalized by biome and realm. | Percentage | CIESIN [55] |
| **Irrigated areas** | Percentage of land equipped for irrigation around the year, 2000. | Percentage | FAO [56] |
| **Land cover and use** | General class of vegetation, tree, and ice cover or purpose of land use, 2020 (resampled and reclassified from Global Land Cover and Land Use) | Built up; cropland; desert; semi-arid; short vegetation; snow or ice; tree cover; wetland | GLAD [57] |
| **Land Surface Temperature** | Interannual averages of daily land surface temperature estimates for daytime, nighttime, and day/nighttime range, 2003–2020. | K | MOD21A1N v006 [58,59] |
| **Nighttime light** | The surface upward radiance from artificial light emissions extracted from at-sensor nighttime radiances at top-of-atmosphere. | $nWatts \cdot cm^{-2} \cdot sr^{-1}$ | NASA Black Marble [60] |
| **Pasture areas** | Proportion of land given over to pasture, 2000. | Proportion | CIESIN [49] |
| **Population density** | Human population density per 1km$^2$. | Inhabitants per km$^2$ | WorldPop [37] |
| **Potential evapotranspiration** | 8-day sum of the water vapor flux under ideal conditions of complete ground cover by plants. | kg/m$^2$/8-day | NASA EOSDIS [61] |
| **Region** | Region of the globe as defined by the World Bank | East Asia & Pacific; Europe & Central Asia; Latin America & the Caribbean; Middle East & North Africa; South Asia | World Bank [40] |
| **Urbanicity** | Urbanicity status at georeferenced location (reclassified from Global Human Settlement database). | Urban; peri-urban; rural; remote | GHS [62] |

**Exploratory spatial data analysis.** We first assessed the presence of spatial autocorrelation by generating semi-variograms of the Pearson residuals from a non-spatial logistic regression that included all explanatory variables listed in Table 2 (S1 Text). We fit spherical spatial correlation models to each semi-variogram and estimated the nugget, range, and sill for each outcome. The semi-variograms and respective models were estimated using the **gstat** R package [41]. Together with the nugget:sill ratio and the estimated range, we determined that an explicitly spatial modeling approach was required to account for the non-trivial spatial correlation in the Pearson residuals.

**Model fitting.** Given the massive spatial scale of the database, with hundreds of thousands of points spanning most of the globe, incorporating spatial correlation into the models presented computational challenges. We used the **inlabru** R package to implement an integrated

nested Laplace approximation (INLA) modeling approach in which all locations are projected onto a coarsened grid or "mesh" containing several thousand vertices that carry the spatial information and can be reprojected onto the observed data [42,43]. INLA models approximate Bayesian models by constructing the posterior distribution and then applying Laplace approximations, thus bypassing the need for time-consuming Markov chain Monte Carlo sampling and making global-scale computation feasible. All coordinates were transformed via the Mollweide projection and scaled into kilometers prior to analysis. The mesh used for modelling had 18,352 vertices, placed within continental boundaries and spatial correlation was specified using a stationary Matérn covariance function. Further details on the implementation of the INLA model are provided in S1 Text.

**Model predictions.** Predicted probabilities for each outcome were made for all locations in the domain of interest (the LMICs) at 0.05 decimal degree$^2$ resolution and exported in Georeferenced Tag Image File format (GeoTIFF). The spatial covariates from Table 2 along with the time variable were used to generate predicted logistic distribution probability of the finished class of each building material from the INLA model. A value for time corresponding to the first of January 2023 was used for making predictions. Missing pixel values were filled by performing imputation using k-Nearest Neighbors method by Python Scikit-learn package [44].

**Model evaluation.** The predictive performance of the spatial models was assessed by calculating common metrics of recall (sensitivity), precision (positive predictive value), accuracy (the proportion correctly classified), F1-score (mean of precision and recall), and area under the receiver operating characteristic curve (ROC-AUC). For each performance metric, two multiclass averaging metrics (macro and weighted average) were calculated, including macro averaging and weighted macro averaging, given by:

$$Pr_{macro} = \frac{1}{n}\sum_{i=1}^{n} Pr_i \tag{1}$$

$$Pr_{weighted-macro} = \frac{1}{n}\sum_{i=1}^{n} Pr_i * Obs_i \tag{2}$$

Where $Pr_i$ is the precision calculated from the multiple class predictions and $Obs_i$ is the number of observations of one class. $n$ is the total number of observations of all classes. To assess the relative contribution of each spatial covariate to the models, feature importance values were approximated by taking the absolute value of the ratio between the mean fixed effect estimates and their standard deviations, a Bayesian analog of the standardized coefficient, which were then plotted for each model.

## Ethics statement

All human subject information used in this analysis was anonymized, publicly available secondary data, and therefore ethical approval was not required or sought. For data provided by the DHS Program, data access requests (including for the displaced cluster coordinates) were submitted and authorized through the Program's website. A completed checklist of Guidelines for Accurate and Transparent Health Estimates Reporting (GATHER [45]) is included in S1 Text.

## Results

350 nationally representative household surveys (together containing data from more than 6 million households in 283,000 clusters) met the inclusion criteria, reported information on

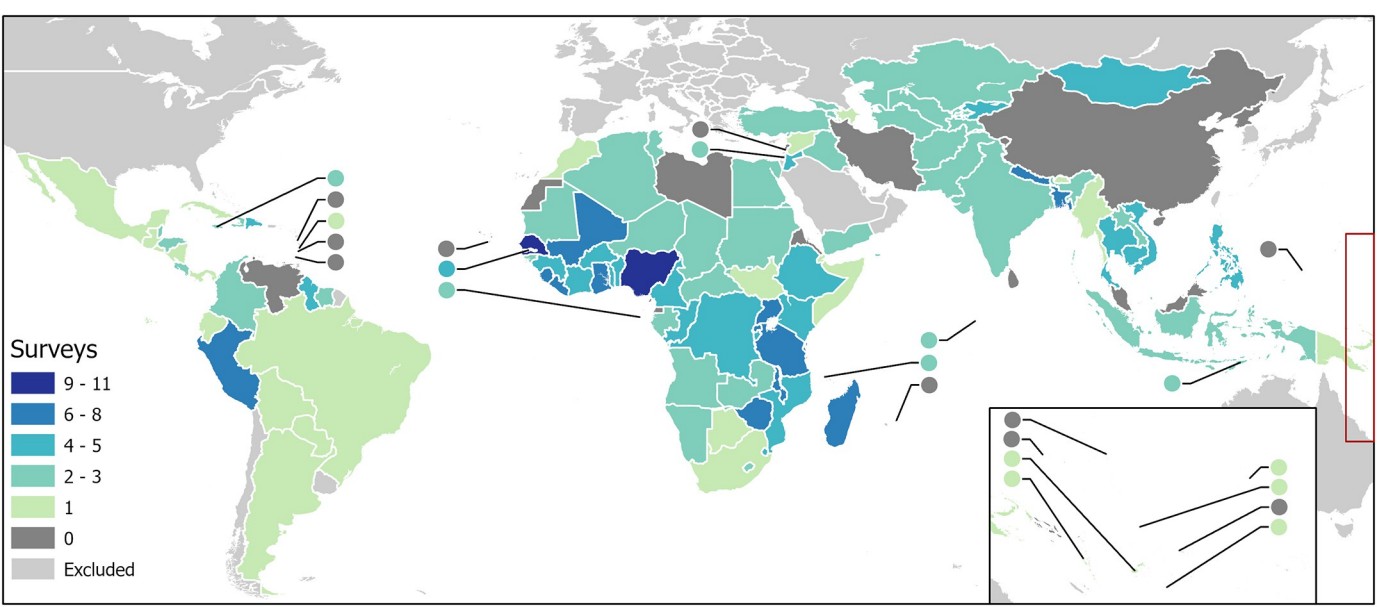

**Fig 1. Number of nationally representative household surveys included in input dataset by country for included LMICs (small countries represented by circles).** Base map compiled from shapefiles obtained from U.S. Department of State—Humanitarian Information Unit [63] and Natural Earth free vector map data @ naturalearthdata.com that are made available in the public domain with no restrictions.

construction material types for one or more of the dwelling components and were included in the model training dataset. Fig 1 shows the number of surveys contributed by each LMIC, while S1 Data gives the national level distribution of each of the three housing construction variables in each survey (before within-cluster sub-sampling, and without sample weights applied). All eligible surveys included information on floor material; however, wall and roof material information were only available from 328 and 324 surveys respectively. No relevant data from household surveys could be found for several LMICs with large geographies and populations, most notably China, Iran, Venezuela, Libya, and Malaysia, as well as the smaller countries of Eritrea, North Korea, Lebanon, Equatorial Guinea, and numerous island nations such as Sri Lanka.

Fig 2 shows the geographical distribution of the coverage of improved materials predicted by the INLA models for each of the three binary dwelling component variables across the domain of included LMICs. These predictions are also provided as raster TIFF files available on the Dryad data repository (https://doi.org/10.5061/dryad.cjsxksnf8). There are some similarities across the variables, with low coverage predicted for all three across a wide belt of the Sahel and southern Sahara regions of Africa, much of inland Amazonia, and areas of the Tibetan plateau, as well as individual countries including the Democratic Republic of the Congo, Mozambique, Madagascar, Pakistan, and Papua New Guinea. High coverage of all three improved components coincided across much of the Middle East, Mediterranean North Africa, the coast of the Bight of Benin, the Caribbean, sub-Amazonian Brazil, southern Argentina, and South Africa. However, divergence in coverage of the three variables is evident over many locations. Across Kazakhstan, Mongolia, Azerbaijan, Cambodia and Laos, low coverage of improved floors, but high coverage of walls and roofs were predicted, while in Afghanistan, the reverse was the case. Yemen has mostly high improved floor coverage predicted, but low improved roof and mixed improved wall coverage, while on the island of Borneo, that pattern

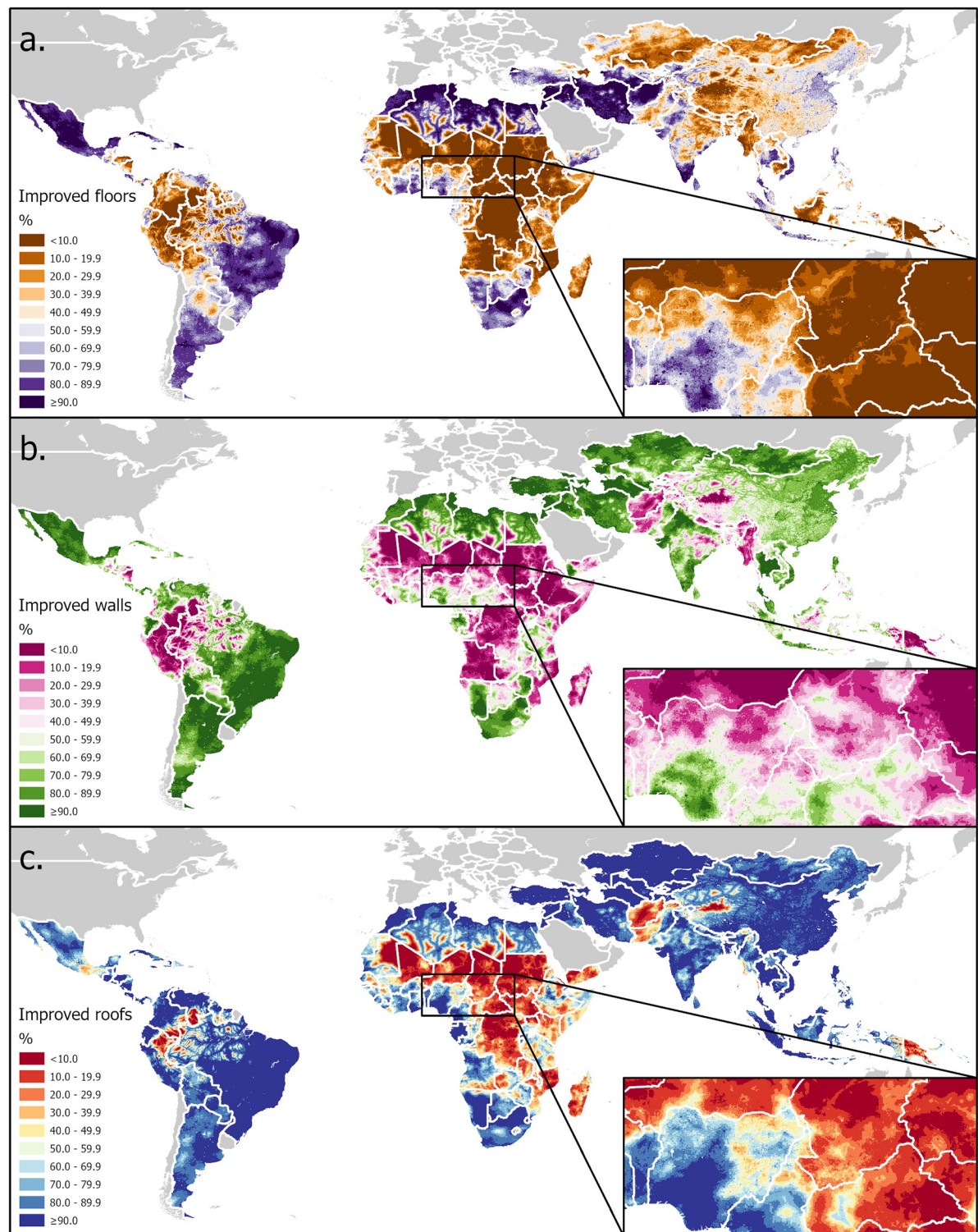

**Fig 2.** Coverage of improved material for three dwelling components—a. floors, b. walls, c. roofs–in LMICs predicted by integrated nested Laplace approximation (INLA) models fitted to household survey data with inset maps showing details at a smaller zoom extent. Base maps compiled from shapefiles obtained from U.S. Department of State—Humanitarian Information Unit [63] and Natural Earth free vector map data @ naturalearthdata.com that are made available in the public domain with no restrictions.

is reversed. Importantly, sub-national patterns are clearly visible, for example, with respect to improved floors, walls, and roofs in India, China, Mexico, and Brazil.

Fig 3 shows ridge plots visualizing the distribution of predicted values for the coverage of improved status for each of the three dwelling components and stratified by the six world regions. The distribution of improved roofs was highly concentrated at values very close to 100% in the Central Asia region, findings which are borne out by the input data, in which most surveys recorded a coverage of finished roofs greater than 97% (S1 Data). This was true to a far lesser extent for other regions—with the exception of Sub-Saharan Africa, which had predicted values much more evenly dispersed along the range of values–and for improved walls, though the South Asia region had a much more dispersed, bimodal distribution for the latter variable. For improved floors, predicted values were highly concentrated at the low extreme of Sub-Saharan Africa.

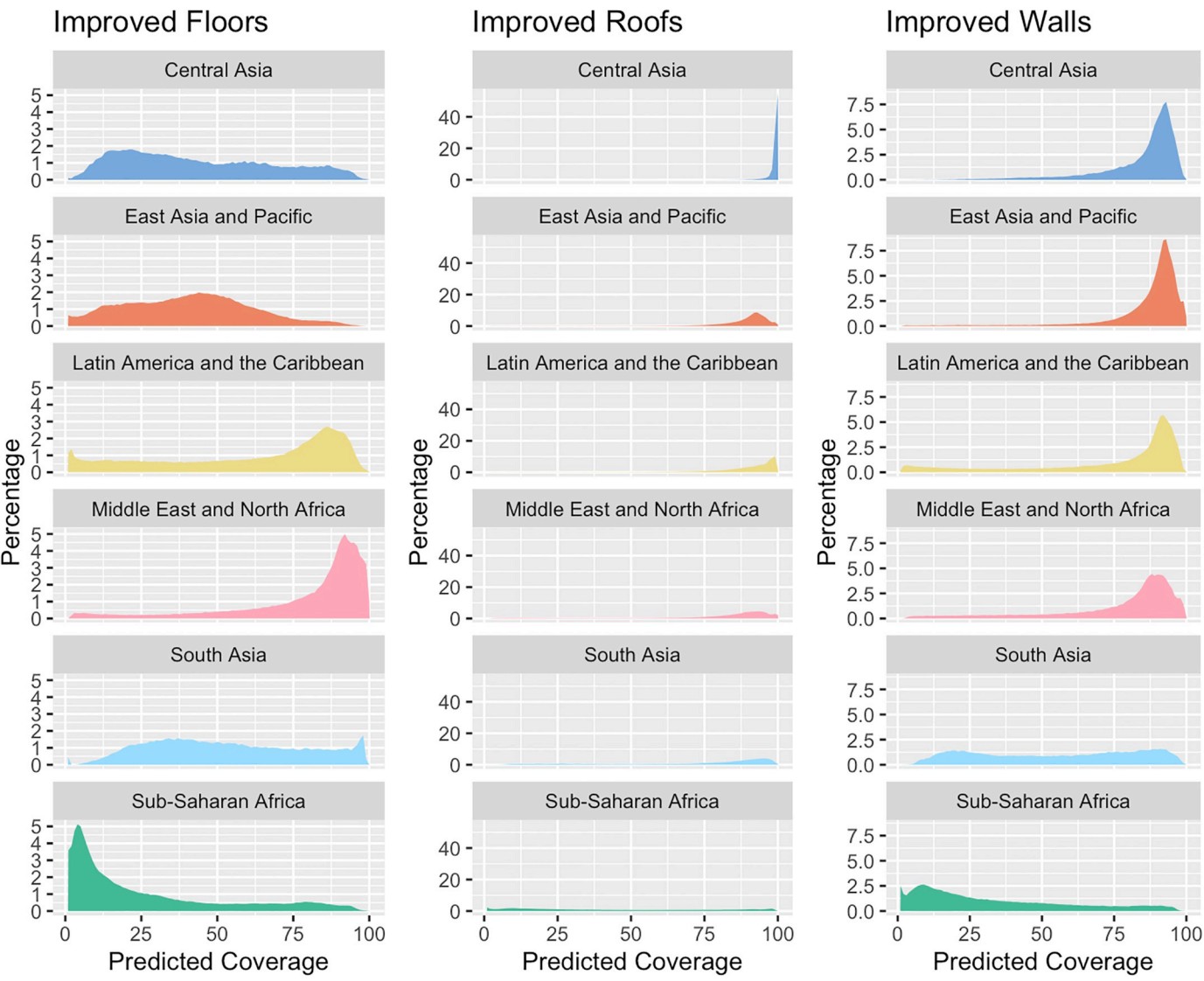

**Fig 3. Distribution of values predicted for coverage of improved dwelling components by INLA models, stratified by component and world region.**

Fig 4 visualizes the feature importance values for each spatial covariate in each of the three models, including for the separate categories of the four factor variables (with time excluded to facilitate comparison between the geographic variables). There were strong similarities in the importance rankings of the variables between the three components, with nighttime light,

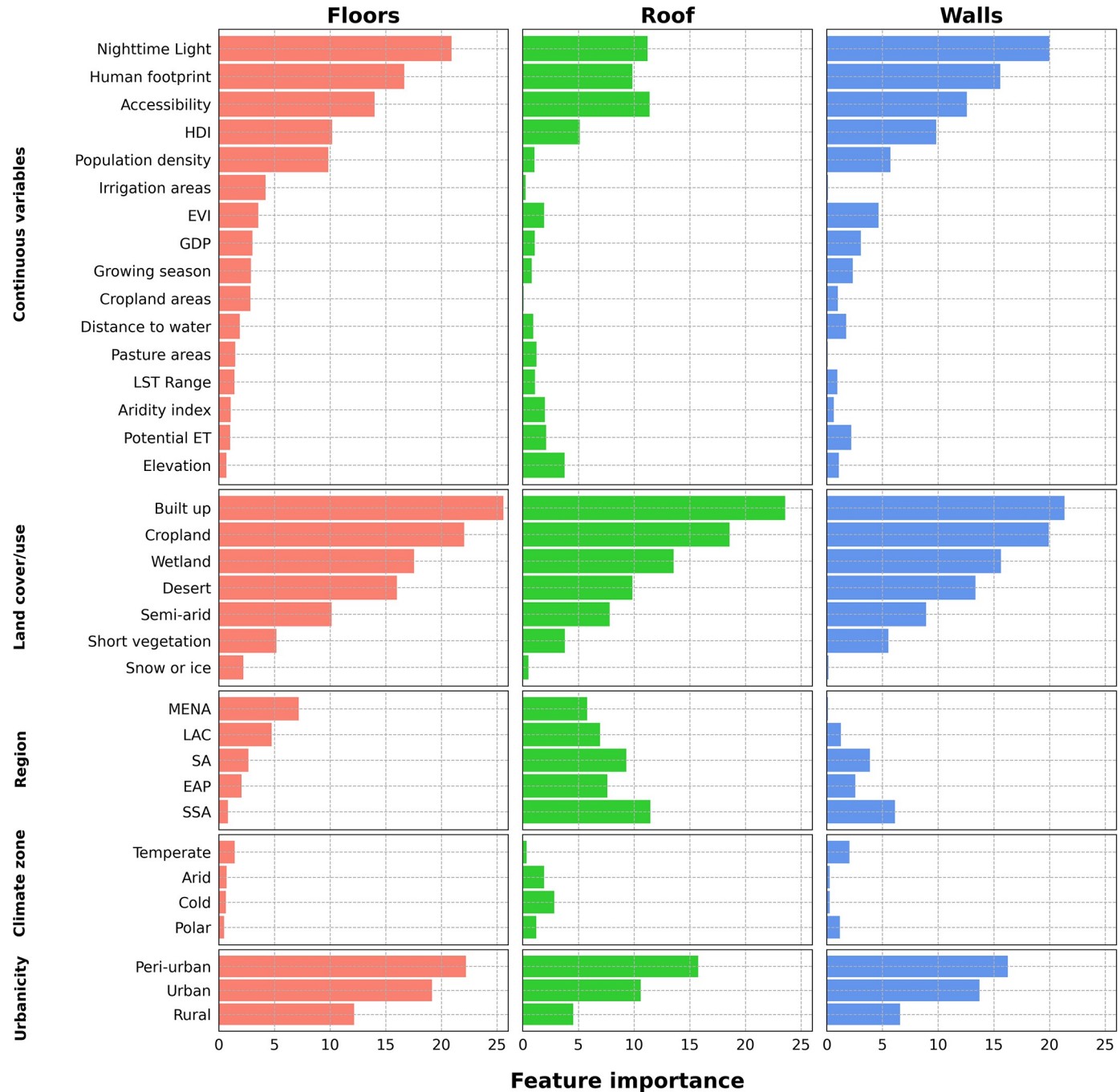

**Fig 4. Feature importance for each of the variables and their categories included in the final model for each of the dwelling components (excluding time.** HDI–Human Development Index; EVI–Enhanced Vegetation Index; LST–Land Surface Temperature; ET–Evapotranspiration; GDP–Gross Domestic Product; MENA–Middle East and North Africa; LAC–Latin America and the Caribbean; SA–South Asia; EAP–East Asia and Pacific; SSA–Sub-Saharan Africa. Comparison categories for factor variables are Land cover/use–tree cover; Climate zone–tropical; Region–Europe and Central Asia; Urbanicity—remote).

**Table 3. Evaluation statistics for models of construction materials for three dwelling components.**

| | | Observations (%) | Precision | Recall | F1-score | ROC-AUC |
|---|---|---|---|---|---|---|
| **Floors** | **Total** | 258,472 (100.0) | - | - | - | 0.85 |
| | **Unimproved** | 108,931 (42.1) | 0.73 | 0.74 | 0.73 | - |
| | **Improved** | 149,541 (57.9) | 0.81 | 0.80 | 0.80 | - |
| | **Macro-average** | - | 0.77 | 0.77 | 0.77 | - |
| | **Weighted average** | - | 0.77 | 0.77 | 0.77 | - |
| **Walls** | **Total** | 248,421 (100.0) | - | - | - | 0.85 |
| | **Unimproved** | 81,621 (32.9) | 0.71 | 0.59 | 0.65 | - |
| | **Improved** | 166,800 (67.1) | 0.82 | 0.88 | 0.85 | - |
| | **Macro-average** | - | 0.77 | 0.74 | 0.75 | - |
| | **Weighted average** | - | 0.78 | 0.79 | 0.78 | - |
| **Roofs** | **Total** | 235,024 (100.0) | - | - | - | 0.87 |
| | **Unimproved** | 46,272 (19.7) | 0.76 | 0.38 | 0.50 | - |
| | **Improved** | 188,752 (80.3) | 0.86 | 0.97 | 0.91 | - |
| | **Macro-average** | - | 0.81 | 0.67 | 0.71 | - |
| | **Weighted average** | - | 0.84 | 0.85 | 0.83 | - |

human footprint and accessibility exhibiting high importance and categories of land cover and use–notably the "built up" and "cropland" classifications–and urbanicity—peri-urban and urban areas–also contributing considerably. Many of the environmental and hydroclimatic variables–notably climate zone, but also EVI, potential evapotranspiration, temperature range, and aridity–showed only modest or negligible feature importance.

Table 3 gives statistics that evaluate the models' performance in classifying household construction material types for the three dwelling components. Across the whole database, floors were the dwelling component for which coverage of improved construction material was lowest at 57.9%, the equivalent coverage for walls and roofs being 67.1% and 80.3% respectively. While precision, recall and F1-score statistics were generally high for the unimproved category in all models, they varied considerably for the improved category, particularly for the roofs model, for which recall, and F1-score were just 0.4 and 0.5 respectively. However, the roofs model was the one with the highest weighted average for those three statistics (a precision of 0.84, recall of 0.85 and F1-score of 0.83, compared with 0.78, 0.79, and 0.78 respectively for the walls and 0.77 for all three statistics for the floors model). All three models demonstrated similarly strong discriminatory power and performance in distinguishing between households with improved and unimproved construction materials in the respective dwelling components, with ROC-AUC statistics of 0.85–0.87.

## Discussion

Housing infrastructure and quality are major determinants of infectious disease risk and other health outcomes, particularly in regions of the world where vector borne, waterborne and neglected tropical diseases are endemic. Although, the nature of this relationship is complex and multifaceted and varies depending on the specific pathogen and vector species, it highlights the importance of targeting interventions to mitigate these adverse health outcomes, particularly in LMICs where the overwhelming majority of childhood mortality occurs. As attention turns to improving housing quality in low-resource settings as a strategy for controlling infectious diseases, it is important to quantify the geographical distribution of

improvements to the major dwelling components to identify and target resources towards populations at risk. This study is the first attempt to meet this objective.

The findings indicate that the use of improved material in housing construction varies markedly by geography and at different scales. Each world region exhibits a distinct distribution of predicted coverage, and within countries themselves, as well as between neighboring countries, the prevalence of improved dwelling components often varies widely (as is visible for Nigeria and central Africa on the inset maps in Fig 2). There are several possible explanations for what drives this variation. It is superficially plausible that differences in the prevailing environmental and climatological conditions from one location to another might determine preferences among the population to and local adaptations of construction methods for the use of particular materials that provide optimal ventilation, insulation or imperviousness to moisture. For example, some populations in Sub-Saharan Africa express a preference for traditional thatched roofs over metal replacements, because they allow for open eaves that provide ventilation during the daytime heat [4,64]. However, none of the climatological variables included in this analysis (temperature range, aridity, evapotranspiration), proved to be important predictors of these outcomes, suggesting that this explanation may be misplaced. Alternatively, in places where certain raw materials are naturally abundant it may make more practical sense to forage for these rather than transporting improved materials from elsewhere. Areas with dense vegetation cover may be more conducive to building with wattle, wicker or thatch. The relative importance of land cover and use, which includes both tree cover and short vegetation as categories, to all three models might lend credence to this explanation, though EVI, a measure of vegetation density, contributed little to any of the models. However, the overwhelming importance of urbanicity and related variables (built up land use, nighttime lights, accessibility) as determinants of material used in all three dwelling components might suggest that a lack of economic integration and market access is the major constraint on further gains in housing improvement. Under this hypothesis the observed variation would be driven by differences in supply of rather than demand for improvements in housing material, with supply in turn constrained by economic connections, resources and infrastructure.

In their study mapping changes in housing in sub-Saharan Africa, Tusting and colleagues found broadly similar patterns to this study for that region using only DHS surveys, with coverage of finished construction materials lowest in central Africa, and highest along the West African coast and in the countries bordering South Africa. However, by extending our analytical domain to include other regions, we find important areas of divergence and differences in distributions between the three dwelling components. This could be due to geographical differences in cultural preferences or policy priorities that determine the order in which resources are invested in the three components as human development improves.

This study is subject to several limitations. Our characterization of housing was constrained by the availability of data from household surveys, which generally only ask about just three components, and don't include questions about other relevant features of the built household environment, such as screens covering openings [65] elevation of sleeping areas or improvements to windows and ventilation [66]. Although the variables were originally in three-class ordinal categorical format, we had to combine categories and model them as dichotomous, because there is currently no way to address adjacent categories and parallel odds using the INLA modeling approach. Additionally, our spatial models assume a stationary (i.e., global) covariance structure that does not vary across the globe. This is likely an oversimplification of the latent spatial effects; however, estimating a non-stationary spatial model at the global scale falls outside the scope of the current article and presents a worthwhile future direction. Likewise, improving the precision of the mesh used by INLA may improve predictions, but with ROC-AUC values already relatively high, this is likely to yield only marginal gains.

Despite these limitations, this study fills an important knowledge gap for targeting geographic areas where housing improvements can be made a priority.. Mosquitoes that transmit malaria (*Anopheles* spp.), dengue (*Aedes* spp.), filariasis and Japanese encephalitis (*Culex* spp.) often enter the home through eaves and other openings [67] and rest on walls and ceilings after ingesting a blood meal (the basis behind indoor residual spraying [IRS] of these surfaces as a malaria control intervention). Indeed, in Africa, 80% of malaria transmission occurs indoors [3] and houses constructed of natural material provide more points of entry [67,68] and preferred resting places [69] for malaria-transmitting mosquitoes, putting housing improvements on the research agendas as potential disease control strategies [66,68]. In rural Gambia, reductions in intradomiciliary mosquito vector abundance and survival were found through installing plywood ceilings [70], closing eaves in thatched roofs [71,72], and replacing thatch with ventilated metal roofing [73]. In rural Uganda, living in a house constructed of traditional materials was associated with increased clinical malaria incidence [74] and parasitemia in children [75] and pregnant women [76], and decreased effectiveness of IRS in reducing *Anopheles* biting rates [75]. Similar protective effects against malaria outcomes have been documented in Burkina Faso [77], Ethiopia [78], Laos [79], Malawi [80], South Africa [81], and Tanzania [82]. Aside from mosquito-borne illnesses, living in households with walls made of mud or thatch carries an increased risk of leishmaniasis infection and indoor abundance of sandfly vectors [83], while in the Americas, Chagas Disease vectors (triatomine bugs) are drawn to houses with thatched palm roofs and mud walls [84]. In a Guatemalan community, for example, the odds of triatomine presence were 3.85 times higher in houses with walls that lacked plastering [85], while in rural Paraguay, an intervention to provide houses with smooth, flat and crack-free walls, reduced triatomine infestation by 96.4%, a comparable effect to that of fumigation [86].

## Conclusions

In conclusion, this study applies a spatially explicit modeling approach to a very large dataset, representative of but standardized across diverse geographies, and collected through rigorous and standardized methodologies. The findings allow us to assess the predictive performance of the models as well as the relative contribution of particular covariate variables, and the resulting predictions are made available in a format that's readily useable by researchers, program planners and other stakeholders (available from https://datadryad.org/stash/dataset/doi:10.5061/dryad.cjsxksnf8). The prevalence of improved roofs and walls is high in the Central Asia, East Asia and Pacific and Latin America and the Caribbean regions, while coverage of improvements in all three components, but most notably floors, is low in Sub-Saharan Africa. The strongest determinants of dwelling component quality, tend to be those relating to urbanization and economic integration, suggesting that housing improvement programs should focus on supply-side interventions that provide the resources for these improvements directly to the populations that need them rather than generating broad based demand. The analytical approach can be repurposed with minimal adaptation for other markers of disease risk measured by household surveys such as water and sanitation access, livestock ownership, and childhood nutrition indicators.

## Supporting information

**S1 Text. Supplementary methods and guideline compliance.**
(PDF)

**S1 Data. National level distribution of each of the three housing construction variables in each survey.**
(XLSX)

**S2 Data. Spatial variation in housing construction material in low- and middle-income countries–raster files of prevalence estimates and standard errors.** Available at https://doi.org/10.5061/dryad.cjsxksnf8.
(ZIP)

## Author Contributions

**Conceptualization:** Josh M. Colston, Pavel Chernyavskiy, Venkataraman Lakshmi, Margaret N. Kosek.

**Data curation:** Josh M. Colston, Bin Fang, Navya Annapareddy.

**Formal analysis:** Bin Fang, Pavel Chernyavskiy, Navya Annapareddy.

**Funding acquisition:** Josh M. Colston, Venkataraman Lakshmi, Margaret N. Kosek.

**Investigation:** Josh M. Colston.

**Methodology:** Josh M. Colston.

**Project administration:** Josh M. Colston.

**Supervision:** Josh M. Colston, Venkataraman Lakshmi, Margaret N. Kosek.

**Visualization:** Bin Fang, Malena K. Nong, Pavel Chernyavskiy.

**Writing – original draft:** Josh M. Colston, Bin Fang, Pavel Chernyavskiy, Margaret N. Kosek.

**Writing – review & editing:** Josh M. Colston, Pavel Chernyavskiy, Venkataraman Lakshmi, Margaret N. Kosek.

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
