## [Decision Letter · Decision Letter 0]

2 Aug 2024

PGPH-D-24-01135

Spatial variation in housing construction material in low- and middle-income countries: a Bayesian spatial prediction model of a key infectious diseases risk factor and social determinant of health

Dear Dr. Colston,

Thank you for submitting your manuscript to PLOS Global Public Health. After careful consideration, we feel that it has merit but does not fully meet PLOS Global Public Health’s publication criteria as it currently stands. Therefore, we invite you to submit a revised version of the manuscript that addresses the points raised during the review process. Please address each reviewer point with a response, indicating revisions to the manuscript that you have made to address reviewer remarks.

We look forward to receiving your revised manuscript.

Kind regards,

Justin V. Remais

Academic Editor

Journal Requirements:

Reviewers' comments:

Reviewer's Responses to Questions

**Comments to the Author**

1. Does this manuscript meet PLOS Global Public Health’s publication criteria? Is the manuscript technically sound, and do the data support the conclusions? The manuscript must describe methodologically and ethically rigorous research with conclusions that are appropriately drawn based on the data presented.

Reviewer #1: Yes

Reviewer #2: Yes

Reviewer #3: Yes

2. Has the statistical analysis been performed appropriately and rigorously?

Reviewer #1: I don't know

Reviewer #2: Yes

Reviewer #3: Yes

3. Have the authors made all data underlying the findings in their manuscript fully available (please refer to the Data Availability Statement at the start of the manuscript PDF file)?

Reviewer #1: Yes

Reviewer #2: Yes

Reviewer #3: Yes

4. Is the manuscript presented in an intelligible fashion and written in standard English?

Reviewer #1: Yes

Reviewer #2: Yes

Reviewer #3: Yes

5. Review Comments to the Author

Reviewer #1: This is a well written and well executed study. The figures and background information and scope of work are clear. The figures are consistent with the purpose of the paper.

However, I recommended minor revision because I see a mismatch between the goals of the paper and the conclusions. I recommend that a paragraph be added to the introduction that explicitly states the scientific advancement of the work. Is it the gathering or presenting of the data? Is it the use of this statistical methods? It is not quite clear to the readers. Once this paragraph is explained, then I believe there may be some restructuring needed to emphasize these points.

For example the conclusion section talks about how your statistical approach is fairly computationally efficient. Then I would like a section describing how it is efficient based on its comparison to other methods, including improved background lit review. If the algorithm is critical in this paper, then more information on the formulation of the MCMC algorithm must be supplied. The current explanation is very thin, and does not give me enough details to weigh the accuracy of it. Broadly, I like the approach, but it could use more information.

Alternatively, if the focus is the implications of the results, then I suggest restructuring the discussion and conclusion sections accordingly.

Reviewer #2: Overall Comments

This paper presents a new model for coverage of improved floor, roof, and wall components in low- and middle-income countries worldwide. This paper expands on previous research looking at housing materials and separates the information into three components. The paper is very well written, offers great background, has rigorous methods, and has well-presented results. My recommendations would be to expand the discussion section to a more wholistic analysis of the data as well as to more explicitly describe the importance of the model and how it could be used in the future.

Specific Comments

Introduction

Can the authors provide brief insight into how the improved model is useful? Can it be used to focus intervention efforts, guide government infrastructure plans, etc.?

Materials and Methods

No comments

Results

There are some countries shown in Figure 2 which are quite close together but have very different coverages, why do you think that may be occurring? What variables from Figure 4 may be driving the prediction? This could be talked about in the discussion.

What are the plot subsets showing in Figure 2?

Discussion

The second paragraph feels like more background/introduction material rather than a discussion.

I would be interested in a more direct discussion of the implication of your results, how they compare to other studies, why they are changing over space, and why the three components may differ from one another.

Are the key drivers of the model from Figure 4 from reliable data sets? Does data reliability or availability vary by country?

Conclusion

The authors state the authors would be in a "useable format", but who is it usable to (future researchers, policy makers, or the general public)?

Reviewer #3: General Comments

This manuscript reports an innovative and robust spatial analysis of global housing quality, making a clear case for the relevance of the analyzed housing characteristics to global public health. The authors have obtained and harmonized a considerable amount of data across many domains to produce subnational-scale maps of predicted housing quality characteristics that will be highly valuable as inputs to future global health analyses, particularly if the corresponding prediction uncertainties can also be mapped and shared.

The methodology is well-described and sound, but additional discussion of a pair of methodological considerations would be beneficial. First, what are the impacts and implications of the two-stage sampling design in the underlying outcome data for the spatial approach employed? While the original survey data are nationally representative, this analysis does not appear to make use of the survey weights; furthermore, a single household was randomly selected to represent each cluster. Can the authors comment on whether the survey design/weights and this within-cluster subsampling are relevant to the spatial models used in this analysis and why or why not that is the case? Second, were temporal aspects considered besides the continuous log-months elapsed variable that was used as model predictor? While a full spatiotemporal analysis would likely be infeasible given the scale of the analysis and that the surveys are not really longitudinal, the assumption that a given housing characteristic changed over time in a globally uniform log-linear manner requires some justification. Have I correctly interpreted the model structure, and was there consideration given to allowing the effects of time to vary by region or other characteristic, or to assessing temporal autocorrelation in the empirical semivariogram analysis?

Specific Comments

L97-99: How many households did each cluster generally contain? Consider reporting e.g. the median and IQR households per cluster to provide context for the proportion of observations withheld from the analysis.

L111: Was there a rationale for using the spherical correlation model? (It’s a perfectly reasonable choice, just curious if there was a specific motivation for choosing it.)

L126: How was the number of mesh vertices selected?

L127-128: Consider noting in the main text that the INLA spatial models were specified with a Matern covariance function.

L132-133: Was prediction uncertainty also assessed? As these prediction maps are intended to be used as inputs for future analyses, it would be useful to also have access to a measure of uncertainty, such as companion maps featuring the prediction standard error and/or upper and lower bounds of the e.g. 90% prediction interval at each location.

L145-149: If feature importance was assessed simply by scaling the regression coefficients, as the text suggests, why could this not be done using the parameter estimates for the fixed effects from the inlabru models? Was some other technique (e.g., permutation feature importance) from Scikit-learn used?

6. PLOS authors have the option to publish the peer review history of their article (what does this mean?). If published, this will include your full peer review and any attached files.

**Do you want your identity to be public for this peer review?** For information about this choice, including consent withdrawal, please see our Privacy Policy.

Reviewer #1: No

Reviewer #2: **Yes: **Claire Anderson

Reviewer #3: No

---

## [Decision Letter · Decision Letter 1]

19 Nov 2024

Spatial variation in housing construction material in low- and middle-income countries: a Bayesian spatial prediction model of a key infectious diseases risk factor and social determinant of health

PGPH-D-24-01135R1

Dear Dr. Colston,

We are pleased to inform you that your manuscript 'Spatial variation in housing construction material in low- and middle-income countries: a Bayesian spatial prediction model of a key infectious diseases risk factor and social determinant of health' has been provisionally accepted for publication in PLOS Global Public Health.

Please consider optional changes recommended by Reviewer 2.

Best regards,

Justin V. Remais

Academic Editor

Reviewer Comments (if any, and for reference):

Reviewer's Responses to Questions

**Comments to the Author**

1. If the authors have adequately addressed your comments raised in a previous round of review and you feel that this manuscript is now acceptable for publication, you may indicate that here to bypass the “Comments to the Author” section, enter your conflict of interest statement in the “Confidential to Editor” section, and submit your "Accept" recommendation.

Reviewer #1: All comments have been addressed

Reviewer #2: All comments have been addressed

Reviewer #3: All comments have been addressed

2. Does this manuscript meet PLOS Global Public Health’s publication criteria? Is the manuscript technically sound, and do the data support the conclusions? The manuscript must describe methodologically and ethically rigorous research with conclusions that are appropriately drawn based on the data presented.

Reviewer #1: Yes

Reviewer #2: Yes

Reviewer #3: Yes

3. Has the statistical analysis been performed appropriately and rigorously?

Reviewer #1: Yes

Reviewer #2: Yes

Reviewer #3: Yes

4. Have the authors made all data underlying the findings in their manuscript fully available (please refer to the Data Availability Statement at the start of the manuscript PDF file)?

Reviewer #1: Yes

Reviewer #2: Yes

Reviewer #3: Yes

5. Is the manuscript presented in an intelligible fashion and written in standard English?

Reviewer #1: Yes

Reviewer #2: Yes

Reviewer #3: Yes

6. Review Comments to the Author

Reviewer #1: Thank you for your response to all of the reviewers' comments. Good job.

Reviewer #2: Overall, this paper is improved in its clarification of methods and discussion. I only have a few minor comments, specified below.

Specific comments (line numbers in reference to the tracked-changes version):

Line 29: Finished floors have also been associated with lower incidences of soil-transmitted helminth infections; the authors may want to add references to that to this statement.

Line 351: I think this paragraph can be made more concise to state that household material improvements has been shown to reduce transmission of vector-born disease, and this study provides context of geographic areas where improvements can be made a priority.

Line 388: At the moment, this link is not functional.

Reviewer #3: (No Response)

7. PLOS authors have the option to publish the peer review history of their article (what does this mean?). If published, this will include your full peer review and any attached files.

**Do you want your identity to be public for this peer review?** For information about this choice, including consent withdrawal, please see our Privacy Policy.

Reviewer #1: No

Reviewer #2: **Yes**

Reviewer #3: No
